# Changes in the Epidemiology of Respiratory Pathogens in Children during the COVID-19 Pandemic

**DOI:** 10.3390/pathogens11121542

**Published:** 2022-12-15

**Authors:** Asmae Lamrani Hanchi, Morad Guennouni, Toufik Ben Houmich, Mohamed Echchakery, Ghizlane Draiss, Noureddine Rada, Said Younous, Mohamed Bouskraoui, Nabila Soraa

**Affiliations:** 1Laboratory of Microbiology, University Hospital Mohamed VI, Faculty of Medicine and Pharmacy, Cadi Ayyad University, Marrakech 40030, Morocco; 2Laboratory of Health Sciences and Technologies, Higher Institute of Health Sciences of Settat, Hassan First University of Settat, Settat 26000, Morocco; 3Pediatric Department, University Hospital Mohamed VI, Faculty of Medicine and Pharmacy of Marrakech, Cadi Ayyad University, Marrakech 40030, Morocco; 4Pediatric Intensive Care Unit, University Hospital Mohamed VI, Faculty of Medicine and Pharmacy of Marrakech, Cadi Ayyad University, Marrakech 40030, Morocco

**Keywords:** SARI, children, respiratory pathogens, mPCR, COVID-19, film array^®^ respiratory panel

## Abstract

Since the outbreak of the COVID-19 pandemic, a significant decrease in non-COVID-19 respiratory illnesses were observed, suggesting that the implementation of measures against COVID-19 affected the transmission of other respiratory pathogens. The aim of this study was to highlight the changes in the epidemiology of respiratory pathogens in children during the COVID-19 pandemic. All children with Severe Acute respiratory illness admitted to the pediatric departments between January 2018 and December 2021 with negative COVID-19 PCR, were enrolled. The detection of respiratory pathogens was made by the Film Array Respiratory Panel. A total of 902 respiratory specimens were tested. A significantly lower positivity rate during the COVID-19 period was found (*p* = 0.006), especially in infants under 6 months (*p* = 0.008). There was a substantial absence of detection of Respiratory Syncytial Virus and Influenza A during the winter season following the outbreak of the pandemic (*p* < 0.05; *p* = 0.002 respectively). An inter-seasonal resurgence of Respiratory Syncytial Virus was noted. Human Rhinovirus was detected throughout the year, and more prevalent in winter during COVID-19 (*p* = 0.0002). These changes could be explained by the impact of the implementation of preventive measures related to the COVID-19 pandemic on the transmission of respiratory pathogens in children.

## 1. Introduction

Severe acute respiratory infections (SARI) are the leading infectious disease in children [1]. It is the most common cause of morbidity and mortality in children worldwide [2,3]. Viral etiology is the most frequent cause of SARI [4,5]. Though, atypical bacteria can also be involved, but in a much smaller proportion [4]. Towards the end of 2019, one more virus has been added to the list of respiratory pathogens. This was the severe acute respiratory syndrome coronavirus 2 (SARS-CoV-2) which causes the COVID-19 (coronavirus disease 2019). In Morocco, the first case of COVID-19, was detected in early March 2020, leading to the setting up of a national response to limit the spread of the virus [6]. Several measures were undertaken including handwashing, social distancing, environmental cleaning, mask wearing and then closure of schools as well as a lockdown. These measures have had a temporary impact on the control of COVID-19 [7].

An unexpected effect on the circulation of other respiratory pathogens has been suggested by several studies [8,9,10]. Moreover a significant decrease in the hospital admission due to non-COVID-19 respiratory infection, particularly those due to respiratory syncytial virus (RSV) and influenza, was observed in both hemispheres at the beginning of the COVID-19 pandemic, suggesting that the implementation of individual and collective measures against COVID-19 affected the transmission of other respiratory pathogens [10,11,12,13]. The aim of this study was to compare the epidemiology of respiratory pathogens detected in children admitted with SARI before and during COVID-19 period and to highlight the changes in the epidemiology of respiratory infections in children in relation to the COVID-19 pandemic.

## 2. Patients and Methods

### 2.1. Patients

This was a descriptive observational and retrospective analysis using data from the microbiology laboratory of the University Hospital MED VI of Marrakech. It covered a period of 4 years, from January 2018 to December 2021. We have divided the study period into two parts: before COVID-19 from January 2018 to February 2020, and during COVID-19 from March 2020 to December 2021.

In the current analysis, we included children under fourteen years of age hospitalized for the management of Severe Acute Respiratory Infection in the Mohamed VI Pediatric University Hospital in Marrakech. While, we excluded children with non-infectious and those who had positive COVID-19 PCR during the COVID-19 period. This is a tertiary pediatric hospital covering the region of Marrakech and receives referrals from the south of Morocco. 

SARI was defined as a diagnosis of severe bronchiolitis, pneumonia, respiratory distress, influenza syndrome in immunocompromised children, or a clinical suspicion of pertussis that required a hospitalization in the pediatric departments. The pediatric inpatients enrolled in this study, were divided into five groups according to age: 0 to 6 months (M), 6 M to 1 year (Y), 1 to 2 Y, 2 to 5 Y and more than 5 Y old. The clinical diagnosis was based on clinical features, medical examination and radiological evidence of SARI when a radiological assessment was required. The respiratory secretions were collected from the pediatric inpatients using flocked nasopharyngeal swabs immersed in a universal transport medium UTM (upper respiratory tract) or bronchial aspiration and bronchoalveolar lavage (lower respiratory tract). [14,15]. The molecular diagnosis was made by multiplex PCR. The Respiratory specimens are processed upon receipt in the microbiology laboratory and results are delivered without delay.

### 2.2. Methods

We used the Film Array^®^ instrument (BioMérieux, Marcy-l’Étoile, France) with the Film Array^®^ Respiratory Panel (FA-RP) for the detection of respiratory pathogens, which simultaneously detects viruses and bacteria in less than one hour. The Film Array is a closed multiplex PCR system that includes extraction, amplification, detection and analysis of samples. FA-RP allows the detection of the following respiratory pathogens: Human Adenovirus (AdV), Human Coronavirus (CoV) 229E, NL63, HKU1 and OC43, Human Metapneumovirus (HMPV), Human Rhinovirus (HRV)/Human Enterovirus (EnV), influenza virus A (Influenza A), A/H1, A/H1-2009 and A/H3 and B (Inf B), Parainfluenza virus 1, 2, 3 and 4 (PIV) and Respiratory Syncytial Virus (RSV). The bacteria included in this panel are *Bordetella pertussis* (Bp), *Chlamydophila pneumoniae* (Chp) and *Mycoplasma pneumonia* (Mp) [16,17].

### 2.3. Statistical Analysis

We used SPSS software (version 23.0; SPSS, Inc., Chicago, IL, USA) and Microsoft Excel (Microsoft Corporation, Washington, USA) to perform statistical analysis. Statistical comparisons were performed using the chi-square test. A probability (*p*) value less than 0.05 was considered statistically significant.

## 3. Results

### 3.1. Paediatric Patient Demographics and Clinical Characteristics before and during COVID-19

A total of 902 specimens from hospitalized children were tested using a Film Array Respiratory Panel over the study period, including 586 samples before the COVID-19 period and 316 during COVID-19. We have not observed any significant difference in age category or gender between the two periods studied. The occurrence of SARI was higher in male children (n = 525, 58.2%). More than half of the patients were less than 6 months old (n = 485, 54.2%), with no significant difference between the two periods before and during COVID-19. The main common reason for hospitalization was severe bronchiolitis with respiratory distress, mostly in children under one year old (*p* < 0.05). Before the COVID-19 period, admissions of children with SARI were higher in winter (n = 252, 43%). While, during COVID-19, it represented only 20.6% (n = 65) and was higher in spring (n = 104, 32.9%) with a statistically significant difference (*p* < 0.001). The Table 1 summarize the demographics and clinical characteristics of patients.

### 3.2. Distribution of Respiratory Pathogens before and during COVID-19

During the study period, a total of 628 (69.6%) of the children were tested positive for at least one respiratory pathogen (*p* = 0.007). The difference between the rate of positivity before COVID-19 (n = 426, 72.7%) and during COVID-19 (n = 202, 63.9%) was statistically significant (*p* = 0.006). This was similar for boys and girls (n = 362/525, 69%; n = 266/377, 70.6%; *p* = 0.6 respectively). The co-infection with more than one pathogen was found in 38.8% (n = 244) of the samples tested. A significant difference comparing the two periods was noted (n = 129/426, 30.28%; n = 115/202, 56.9%; *p* < 0.001). The predominance of viral etiology in SARI was found both before and during COVID-19. The main viruses detected were HRV (n = 338), RSV (n = 214), followed by PIV (n = 63), Influenza A (n = 40), Coronavirus (n = 38), Human Metapneumovirus (n = 33). Nevertheless, Bacteria were found in 7.6% (n = 48), and the most detected was *Bordetella pertussis* (n = 32). Quantitatively a substantial difference was observed for Influenza A virus, Human Metapneumovirus and *Bordetella pertussis* before and during the COVID-19 outbreak (*p* = 0.002; *p* = 0.039; *p* = 0.0001 respectively) (Table 2). These respiratory pathogens were less detected during COVID-19 than before. Human Rhinovirus was more prevalent during COVID-19 but without significant difference.

### 3.3. Age Distribution of Severe Acute Respiratory Infections before and during COVID-19

Among all included patients, there was a statistically significant difference in the positivity rate according to age group (*p* = 0.007). In children under 5 years old, it was significantly higher than in children over 5 years old *p* < 0.05. A remarkable difference was observed between the two periods in the under six-month age group in which the positivity rate was lower during the COVID-19 period (*p* = 0.008) (Table 3).

### 3.4. Seasonality

Overall, a statistically significant difference in the seasonal variation of the respiratory pathogen detection rate was observed (*p* < 0.001). It was more common in winter (n = 253, 79.8%) followed by autumn (n = 118, 69%), spring (n = 173, 67.8%) and summer (n = 84, 52%). A comparison between the two studied periods showed remarkable differences (*p* = 0.006). HRV was detected throughout the year in the two periods. However, it was statistically more prevalent in winter during COVID-19 (*p* = 0.0002). Concerning RSV, its detection was more frequent in winter (*p* < 0.0001) before COVID-19. whereas, after COVID-19 its detection was found throughout all seasons with a statistically significant difference in autumn, winter and summer (*p* = 0.0001; *p* = 0.0001; *p* = 0.0001 respectively). For a more relevant analysis we studied the annual variations of the monthly distribution of RSV. It showed a disappearance of RSV after the beginning of the pandemic in March 2020, with no detection in the following winter season. The reappearance of RSV was found in April 2021 and showed a peak in June 2021 and another peak in November 2021 (Figure 1). These variations were statistically significant for the following months: January, February, March, June, July, September, October, November, December (*p* < 0.05). Concerning Influenza A, a statistically significant difference was observed in its seasonal distribution (*p* = 0.002). Moreover, the detection of Influenza A was higher in winter before COVID-19. It disappeared after the outbreak of the pandemic and did not re-emerge until December 2021 (Figure 1). For the PIV, it was statistically more prevalent in autumn before COVID-19. As for the period during COVID-19, it was rather found in winter and spring. As for Coronavirus, it was more prevalent in winter during COVID-19 (*p* = 0.03). Concerning *Bordetella pertussis*, it was significantly lower during COVID-19 (*p* < 0.001) (Figure 1). More details are shown in Table 4.

## 4. Discussion

Through this study, we analyzed and compared the epidemiology of respiratory pathogens involving SARI in children before and during the COVID-19 pandemic in order to identify the possible impact of SARS-CoV-2 and the preventive measures implemented since the beginning of the COVID-19 pandemic on the respiratory pathogens responsible for SARI. To our knowledge, this is the first study in Morocco providing a comparative analysis of changes in the epidemiology of respiratory pathogens in children during COVID-19. 

Since the late 2019, SARS-CoV-2 was listed as a novel respiratory pathogen that caused COVID-19 19. According to the World Health Organization more than 549 Million cases of COVID-19 infections have been reported. The number of recorded deaths was 6.34 million until the first of July [18,19]. Nevertheless, COVID-19 disease was not considered particularly serious in children [20]. The first Moroccan case was detected in early March 2020 imposing a general lockdown from 20 March [6].

SARI, especially severe bronchiolitis, are one of the main reasons of hospitalization of infants with a predictable seasonality (Taylor and Whittaker 2022) [4,21]. However, many authors have reported a remarkable decrease in respiratory infections especially during the winter after the outbreak of COVID-19 [9,22,23,24]. This was also the case in our study, which noted a significant decrease in the cohort of children with SARI requiring multiplex PCR testing especially in winter. Additionally observed was the significantly lower positivity rate during the COVID-19 period especially in infants under 6 months. Fourgeaud et al. [10], found in their study a higher median age of children admitted for RSV-associated SARI during the 2020/2021 epidemic (6–11 months instead of less than 6 months in previous epidemics). This could be due to an increase in the RSV-naive cohort and could also explain the decrease in severe infections and intensive care unit admissions in 2020/2021 compared to previous epidemics. 

A significantly higher number of co-infections with multiple viral pathogens was observed especially during COVID-19. However, we have no idea whether the co-infection with multiple respiratory pathogens were more severe due to the lack of clinical characteristics or clinical outcome.

Consistently, in our study, HRV/EV, RSV, PIV, INFLUENZA A and COV were the most identified, confirming the predominance of viral etiology of SARI in children [4]. However, during COVID-19 INFLUENZA A; HMPV and Bp were remarkably less detected. For RSV, it seems that there were no significant differences in the overall number between the two periods. However, its distribution changed significantly initially after the outbreak of COVID-19, and was seasonal before COVID-19 with a peak in the winter. Whereas during the COVID-19 pandemic, there was a loss of the seasonal character of RSV infection with a more remarkable detection in summer and autumn but much less in winter.

Moreover, our study revealed a disappearance of RSV during the winter of 2020–2021 following the COVID-19 pandemic compared to the winters of 2018–2019 and 2019–2020. The resurgence of RSV did not occur until April 2021 with a first peak in the summer, which was not observed in previous years. A second peak occurred in November, signaling the beginning of the winter epidemic of 2021–2022. 

The influenza virus was also affected by changes in its distribution which usually circulated during the winter season before COVID-19 and disappeared during the pandemic and only re-emerged in December 2021. The same finding was reported for *Bordetella pertussis* which was not detected during COVID-19. The same trend was reported by Yeoh et al. [25] during the 2020 Australian winter, which reported a 98.0% to 99.4% decrease in the detection of RSV and influenza infections, respectively, compared to the previous winter seasons of 2012 to 2019. 

This finding suggests that the barrier measures undertaken in morocco to limit the spread of SARS-CoV-2, in particular: lock down, barrier measures, obligation of face masks, school closures, as well as a curfew and the closure of the air space limiting national and international traffic, would have an impact on the circulation of respiratory viruses, in particular influenza and RSV. Indeed, the decrease in respiratory infections with INFLUENZA A and RSV has been attributed to the implementation of local restrictions of COVID-19 by several authors [10,13,26,27]. Nevertheless, even after the sequential relaxation of restrictions related to COVID-19 19 and the reopening of schools in October 2020 with the barrier measures and the restrictions on international travel maintained, the RSV did not circulate in winter 2020–2021. This being consistent with the hypothesis of potential impact of adults in RSV transmission [28]. The same observation was reported by Yeoh et al. [25], who suggested that international, and to a minor extent, national border restrictions may have had an important impact on the circulation of these viruses by preventing their introduction from elsewhere. 

The inter-seasonal resurgence of RSV (until April 2021) suggests that off-season reservoirs of RSV in adults play an important role in the transmission chains of RSV in the population. Factors that may influence this inter-seasonal resurgence include virus introduction (or lack thereof), RSV dynamics in other countries, seasons, climate, and movement of travelers [10,28].

Our study emphasized that, unlike RSV, Influenza virus was more sensitive to barrier measures and did not reappear until December 2021. This may suggest that increased barrier measures, particularly in the vulnerable population, could significantly reduce the incidence of Influenza A infections during the winter season.

Other viruses such as Rhinovirus was less affected by the outbreak of COVID-19 which was detected throughout the year. However, it was statistically more prevalent in winter during COVID-19 (*p* = 0.0002). This could be explained by the virologic characteristics of Rhinovirus [29]. Huh et al. [26] in their study suggested that the persistence of Rhinovirus detection could be explained by the increased exposure of vulnerable children in schools and a low cross-immunity to heterogeneous Rhinovirus subtypes. As for Coronavirus, it was more prevalent in winter during COVID-19. It looks like Rhinovirus and Coronavirus have replaced seasonal viruses during COVID-19.

### Limitations of the Work

The limitations of this study include a lack of information regarding clinical and biological characteristics, management, as well as clinical outcome of patients.

## 5. Conclusions

A significant decrease in the prevalence of respiratory pathogens was observed after the outbreak of COVID-19, in particular RSV and Influenza A as well as *Bordetella pertussis*. This could be explained by the impact of the implementation of preventive measures related to the COVID-19 pandemic in the transmission of respiratory pathogens in children.

## Figures and Tables

**Figure 1 pathogens-11-01542-f001:**
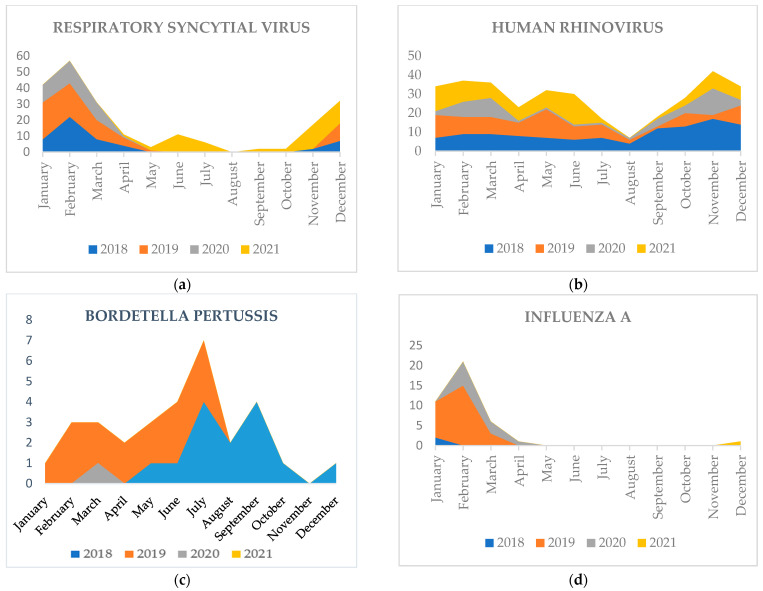
Monthly distribution of RSV (**a**), INFLUENZA A (**b**), HRV (**c**), Bp (**d**).

**Table 1 pathogens-11-01542-t001:** Pediatric Patient demographics and clinical characteristics before and during COVID-19 (n = 902).

	Before COVID-19 (N: 586)	During COVID-19 (N: 316)	Total (N: 902)	*p* Value
Male gender	340 (58%)	185 (58.5%)	525 (58.2%)	0.4
Female gender	246 (42%)	131 (41.5%)	378 (41.8%)	
Age in months (mean/median)			17.5 (4)	
Age range				
<6 months	329 (56.1%)	156 (50.7%)	485 (54.2%)	0.1
6 months to 1 year	120 (20.5%)	61 (19.7%)	181 (20.2%)	
1 to 2 years	47 (8.0%)	39 (12.6%)	86 (9.6%)	
2 to 5 years	40 (6.8%)	28 (9.1%)	68 (7.6%)	
>5 years	50 (8.5%)	25 (8.1%)	75 (8.4%)	
Clinical features				
Bronchiolitis	121 (20.6%)	65 (20.6%)	186 (20.6%)	0.9
Severe Bronchiolitis with respiratory distress	260 (44.4%)	161 (50,9%)	421 (46.7%)	0.059
Pneumonia	95 (16.2%)	38 (12.0%)	133 (14.7%)	0.09
Suspicion of Pertussis	125 (21.3%)	27 (8.5%)	152 (16.9%)	**0.001**
Influenza in immunocompromised children	15 (2.6%)	5 (1.6%)	20 (2.2%)	0.3
Other	48 (8.2%)	77 (24.4%)	125 (13.9%)	
Season of admission				
Autumn	104 (17.7%)	67 (21.2%)	171 (19%)	
Winter	252 (43.0%)	65 (20.6%)	317 (35.1%)	**<0.001**
Spring	151 (25.8%)	104 (32.9%)	255 (28.3%)	
Summer	79 (13.5%)	80 (25.3%)	159 (17.6%)	

**Table 2 pathogens-11-01542-t002:** Distribution of respiratory pathogens before and during COVID-19.

	Total N (%)	Before COVID-19 N (%)	During COVID-19 N (%)	*p*-Value
Human Rhinovirus	338 (53.8)	211 (36)	127 (40.2)	0.2
Respiratory Syncytial Virus	214 (34.1)	149 (25.4)	65 (20.6)	0.1
Parainfluenza	63 (10)	36 (6.1)	27 (8.5)	0.17
Influenza A	40 (6.4)	35 (6)	5 (1.6)	**0.002**
Coronavirus	38 (6.1)	23 (3.9)	15 (4.7)	0.5
Human Metapneumovirus	33 (5.3)	27 (4.6)	6 (1.9)	0.039
Adenovirus	33 (5.3)	23 (3.9)	10 (3.2)	0.5
Influenza B	6 (1)	6 (1)	0 (0)	0.07
*Bordetella pertussis*	32 (5.1)	31 (5.3)	1 (0.3)	**<0.001**
Chlamidophila pneumoniae	2 (0.2)	2 (0.3)	0 (0)	0.2
*Mycoplasma pneumoniae*	14 (1.6)	14 (2.4)	0 (0)	**0.06**
Total	628 (69.6)	426 (72.7)	202 (63.9)	**0.006**

Parainfluenza includes piv1 piv2 piv3 and piv4. CoV includes Cov HKU1, Cov NL63, Cov OC43, Cov 229E. Influenza A includes Influenza A/H1-2009, Influenza A/H1, Influenza A/H3.

**Table 3 pathogens-11-01542-t003:** Prevalence of respiratory pathogens in the five age groups (n = 895).

	Total No of Samples	Total No of Positive Samples (%)	No of Positive Samples (%) before COVID-19	No of Positive Samples (%) during COVID-19	*p*-Value
[<6 months]	485	346 (71.3)	247 (75.1%)	99 (63.5%)	**0.008**
[6 months–1 an]	181	129 (71.3)	87 (72.5%)	42 (68.9%)	0.6
[1–2 years]	86	60 (69.8)	35 (74.5%)	25 (64.1%)	0.2
[2–5 years]	68	50 (73.5)	32 (80%)	18 (64.3%)	0.1
[>5 years]	75	38 (50.7) **	25 (50%)	13 (52%)	0.8
Total	895	623 (69.6) *	429 (72.7)	197 (63.8)	**0.006**

* All group: *p* = 0.007. ** Compared with <5 years: *p* < 0.05.

**Table 4 pathogens-11-01542-t004:** Prevalence of respiratory pathogens in seasons before and during COVID-19.

		Autumn N (%)	Winter N (%)	Spring N (%)	Summer N (%)	Total N (%)	*p*-Value a
Human Rhinovirus	Before COVID-19	**52 (50)**	71 (28.2)	55 (36.4)	33 (41.8)	211 (36)	**0.001**
	During COVID-19	**36 (53.7)**	**34 (52.3)**	36 (34.6)	21 (26.3)	127 (40.2)	**0.001**
	*p* value b	0.6	**0.0002**	0.7	**0.03**	**0.2**	
Respiratory Syncytial Virus	Before COVID-19	2 (1.9)	**117 (46.4)**	30 (19.9)	0	149 (25.4)	**<0.0001**
	During COVID-19	19 (28.4)	14(21.5)	15 (14.4)	17 (21.3)	65 (20.6)	**<0.0001**
	*p* value b	**0.0001**	**0.0001**	0.2	**0.0001**	0.1	
Parainfluenza	Before COVID-19	**14 (13.5)**	11 (4.4)	6 (4)	5(6.3)	36 (6.1)	**0.006**
	During COVID-19	2 (3)	**7 (10.8)**	**16 (15.4)**	2(2.5)	27 (8.5)	**0.006**
	*p* value b	**0.02**	**0.04**	**0.001**	0.2	0.1	
Influenza A	Before COVID-19	0	**32 (12.7)**	3 (2)	0	35 (6)	**<0.0001**
	During COVID-19	0	1 (1.5)	4 (3.8)	0	5 (1.6)	0.1
	*p* value b		**0.009**	0.3		**0.002**	
Human Metapneumovirus	Before COVID-19	0	12 (4.8)	**14 (9.3)**	1 (1.3)	27 (4.6)	**0.002**
	During COVID-19	1 (1.5)	2 (3.1)	2 (1.9)	1 (1.3)	6 (1.3)	0.8
	*p* value b	0.2	0.5	**0.01**	0.9	**0.039**	
Adenovirus	Before COVID-19	3 (2.9)	4 (1.6)	**9 (6)**	**7 (8.9)**	23 (3.9)	**0.013**
	During COVID-19	2 (3)	2 (3.1)	4 (3.8)	2 (2.5)	10 (3.2)	0.9
	*p* value b	0.9	0.4	0.4	0.08	0.5	
Coronavirus	Before COVID-19	6 (5.8)	6 (2.4)	6 (4)	5 (6.3)	23 (3.9)	0.2
	During COVID-19	9 (4.5)	11 (7.7)	13 (6.7)	5 (0)	38 (4.7)	0.1
	*p* value b	0.7	**0.03**	0.3	**0.02**	0.5	
Influenza B	Before COVID-19	0	**6 (2.4)**	0	0	6 (1)	**0.04**
	During COVID-19	0	0	0	0	0	
	*p* value b		0.2			**0.07**	
Bordetella pertussis	Before COVID-19	5 (4.8)	5(2)	8(5.3)	**13 (16.5)**	31 (5.3)	**0.00014**
	During COVID-19	0	0	1	0	1	
	*p* value b	0.06	0.2	0.06	**0.0001**	**0.0001**	

Parainfluenza includes piv1, piv2, piv3 and piv4. Coronavirus includes Cov HKU1, Cov NL63, Cov OC43, Cov 229E. Influenza A includes Influenza A/H1-2009, Influenza A/H1, Influenza A/H3. Autumn: September–October–November; Winter: December–January–February; Spring: March–April–May; Summer: June–July–August. *p* value a: Comparison in the period. *p* value b: Comparison between the two periods before and durig COVID-19.

## Data Availability

The Excel and SPSS file data used to support the findings of this study are available from the corresponding author upon request.

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
