# Peer review of "Changes in the Epidemiology of Respiratory Pathogens in Children during the COVID-19 Pandemic"

_pathogens, 2022, doi:10.3390/pathogens11121542_

Round 1

Reviewer 1 Report

Congratulations to our colleagues in Morocco for this article. It provides valid data about the shift of respiratory viral pathogens during the pandemic. However the article needs a major revision for the English language and the content flow. Please rewrite with good sentence flow to improve the enthusiasm for the readers. One example is line 84 the study says "retro-prospective" when it should say retrospective. There multiple grammatical and sentence formation errors which should be taken into account and rectified prior to acceptance. 

Author Response

Response to Reviewer 1 Comments

We do thank you for goving us the opportunuty of correcting this manuscript. As recommended we were able to make the required changes and correct some issues in different sections of the manuscript.

Point 1: The article needs a major revision for the English language and the content flow. Please rewrite with good sentence flow to improve the enthusiasm for the readers. One example is line 84 the study says "retro-prospective" when it should say retrospective. There multiple grammatical and sentence formation errors which should be taken into account and rectified prior to acceptance.

Response 1: We do thank you very much for your consideration of this manuscript. The manuscript has been proofread and corrected by a native English speaker.

Point 2: The conclusions are not supported by the results

Response 2: The conclusions have been totally rectified and reformulated.

Reviewer 2 Report

Review report

The objective of this study was to observe the changing epidemiology of respiratory pathogens in children during the Covid-19 pandemic in two-time periods 1) 03/2020 to 12/2021 (Case interval Ncase=316), and   2) 01/2018 to 02/2020 (Control interval Ncontrol=586) and Using Film Array Respiratory Panel. Their Film Array, PCR system detects more than 10 viral and 4 common bacterial respiratory pathogens. They evaluated total N=902 specimens and concluded that implementation of preventive barrier measures in adult could have an impact on the transmission of respiratory pathogens in children, especially RSV and INF A infections.

The authors have done a remarkable job in this article, the style is flawless, language clear and concise. The paper is robust in scientific facts and biostatistical analytical sections. The paper is also well organized and ideas clearly expressed. Very beautiful graphs are also included to nicely illustrate their concepts. This is a first-of-its-kind study in Morocco, which must be appreciated and accepted. The study was cleared and approved by Ethics Committee which adds credence to their work.

Suggestions

1. Line 3 —Title change “Covid” to “Covid-19”
2. In Many places in the article the word “Covid”(line 51, line 3, Line 55) or “Covid 19”  (line 44, line 66) is mentioned instead of Covid-19 (line 71), this is erroneous as plain “Covid” could mean SARS1 (SARS COVID-19 Hong Kong outbreak), please change every “Covid”/ “Covid 19” to “Covid-19” to avoid ambiguity. Another option is to use “COVID-2” or “SARS Covid-2”(even this is confusing)
3. Line 130 —“represented only 20,6%” – correct to? “Represented only 20.6%”
4. Line 54 — “An inter-seasonal resurgence of RSV was noted. HRV was detected throughout the year,” — please expand RSV, HRV to Respiratory Syncytial Virus so on
5. Line 57— “especially RSV and INF A infections. This will reduce the burden on pediatric hospitals.”— please expand RSV, INF-A to Respiratory Syncytial Virus so on
6. Lines 307, 289— University Cadi Ayyad (N° 22/2021). Is N° a typographical error? Do you mean No: 22/2021?

1. In contrast, the methodology is biased, in that attempting to observe changes in children who rarely became affected by Covid-19, hence the result was naturally zero and almost like a baseline with RSV, INF A, the universal and most common culprits(=baseline pre-covid-19 era).  It is divided (expected) zero by (baseline) zero study so the result is also ZERO.
2. Limited (N=4) bacterial pathogens were studied, in the community, bacteria account for 30 to 70% of Severe Acute Respiratory infections. Their panel had only 4 common bacteria.
3. All groups of children babies, toddlers, and school-going children were combined into a single group for analysis, which is erroneous as disease patterns are different in different age groups. A stratified analysis would have been better.
4. Clinical characteristics of patients were not included which would have made this study more interesting.
5. In terms of novelty, this paper contributes nothing new.

Author Response

Response to Reviewer 2 Comments

We do thank you for goving us the opportunuty of correcting this manuscript. As recommended we were able to make the required changes and correct some issues in different sections of the manuscript.

Point 1: English language and style are fine/minor spell check required

Response 1: We do thank you very much for your consideration of this manuscript. It been proofread and corrected by a native English speaker.

Point 2:

  1. Line 3 Title change “Covid” to “Covid-19”
    2. In Many places in the article the word “Covid”(line 51, line 3, Line 55) or “Covid 19”  (line 44, line 66) is mentioned instead of Covid-19 (line 71), this is erroneous as plain “Covid” could mean SARS1 (SARS COVID-19 Hong Kong outbreak), please change every “Covid”/ “Covid 19” to “Covid-19” to avoid ambiguity. Another option is to use “COVID-2” or “SARS Covid-2”(even this is confusing)
    3. Line 130 —“represented only 20,6%” – correct to? “Represented only 20.6%”
    4. Line 54 — “An inter-seasonal resurgence of RSV was noted. HRV was detected throughout the year,” — please expand RSV, HRV to Respiratory Syncytial Virus so on
    5. Line 57— “especially RSV and INF A infections. This will reduce the burden on pediatric hospitals.”— please expand RSV, INF-A to Respiratory Syncytial Virus so on

Response 2: all the modifications suggested have been made.

Comment 1: In contrast, the methodology is biased, in that attempting to observe changes in children who rarely became affected by Covid-19, hence the result was naturally zero and almost like a baseline with RSV, INF A, the universal and most common culprits(=baseline pre-covid-19 era).  It is divided (expected) zero by (baseline) zero study so the result is also ZERO.

Response 3: The children were indeed less affected by Covid-19. We excluded all children who tested positive for Covid-19 from this study. We were interested in changes in the circulation of other respiratory pathogens involved in severe acute respiratory infections in children during the Covid-19 pandemic.

Comment 2: Limited (N=4) bacterial pathogens were studied, in the community, bacteria account for 30 to 70% of Severe Acute Respiratory infections. Their panel had only 4 common bacteria.

Response 4: The panel used only detects viruses and atypical bacteria. other bacteria are not detected. According to the clinical caracteritics, in particular severe bronchiolitis with or without respiratory distress, the viral aetiology is predominant. In the case of a suspected acute low respiratory infection, the pneumoniae panel may be used to complete the test, as it looks for more bacteria such as streptococcus pneumoniae.

Comment 3: All groups of children babies, toddlers, and school-going children were combined into a single group for analysis, which is erroneous as disease patterns are different in different age groups. A stratified analysis would have been better.

Response 5: infants less than 6 months old constitute more than 50% of the study population and more than 70% of the children included in this study are less than two years old. Other age categories are less represented.

Comment 4: Clinical characteristics of patients were not included which would have made this study more interesting

Response 6: The limitations of this study include a lack of information regarding clinical and bio-logical characteristics, management, as well as clinical outcome of patients.

The data were collected from the microbiology laboratory. Other studies are in progress including clinical and biological data as well as the outcome of the patients.

Comment 5: In terms of novelty, this paper contributes nothing new

Response 7: To our knowledge this is the first study carried out in morocco. It is important to study the local epidemiology in order to manage the patients better

Round 2

Reviewer 1 Report

Congratulations ! Very well edited and all the feedbacks were incorporated.

Only suggestion would be to use Influenza A instead of Inf A wherever the illness was mentioned in the article. 

Author Response

Thank you very much for your consideration of this manuscript.

The requested suggestions have been corrected on the manuscript.